# Circadian Rhythms and Measures of CNS/Autonomic Interaction

**DOI:** 10.3390/ijerph16132336

**Published:** 2019-07-02

**Authors:** Francesco Riganello, Valeria Prada, Andres Soddu, Carol di Perri, Walter G. Sannita

**Affiliations:** 1Coma Science Group, GIGA-Consciousness, GIGA Institute, University Hospital of Liège, 4000 Liège, Belgium; 2Department of Neuroscience, Rehabilitation, Ophthalmology, Genetics and Maternal/Child Sciences, University of Genova, Polyclinic Hospital San Martino IRCCS, 16132 Genova, Italy; 3Department of Physics and Astronomy, Brain and Mind Institute, The University of Western Ontario, London, ON N6A 3K7, Canada; 4Centre for Clinical Brain Sciences, The University of Edinburgh, Edinburgh, EH16 4SB, UK

**Keywords:** circadian rhythms, sleep-wakefulness cycle, autonomic system, central/autonomic systems interaction, HRV, heart rate variability

## Abstract

The physiological role and relevance of the mechanisms sustaining circadian rhythms have been acknowledged. Abnormalities of the circadian and/or sleep-wakefulness cycles can result in major metabolic disorders or behavioral/professional inadequacies and stand as independent risk factors for metabolic, psychiatric, and cerebrovascular disorders and early markers of disease. Neuroimaging and clinical evidence have documented functional interactions between autonomic (ANS) and CNS structures that are described by a concept model (Central Autonomic Network) based on the brain-heart two-way interplay. The circadian rhythms of autonomic function, ANS-mediated processes, and ANS/CNS interaction appear to be sources of variability adding to a variety of environmental factors, and may become crucial when considering the ANS major role in internal environment constancy and adaptation that are fundamental to homeostasis. The CNS/ANS interaction has not yet obtained full attention and systematic investigation remains overdue.

## 1. Introduction

Circadian rhythms are governed by a biological *master clock* (synchronized on the fluctuations in light intensity and temperature) and by intrinsically cyclic *clock genes* regulated by the hormonal status and environmental factors. This arrangement controls metabolism, endocrine secretion, cardiovascular and motor activity, and the sleep-wakefulness cycle of light-sensitive organisms, thus guaranteeing homeostatic constancy, efficiency of physiological processes, and the adaptation to internal/external changes and requirements [1,2,3,4,5,6,7,8,9,10,11,12]. The suprachiasmatic nuclei (SCN) serve as the central pacemaker driven by information from the retino-hypothalamic tract and by signals from neuronal networks and molecules selectively crossing the blood-brain barrier [4,13,14,15,16]. Adaptation and behavior are influenced by the SCN via neuronal and neuroendocrine cues and synchronization of the peripheral clocks in organs and tissues [17,18] that also integrate information from the environment to coordinate appropriate responses [19]. The circadian system sets time windows, particularly in the alternation of sleep and wakefulness, that often reach pathophysiological relevance [20,21,22,23]. There is full evidence of a complex interaction between the central (CNS) and autonomic (ANS) nervous systems, with a two-way modulation of, and by, circadian rhythms.

## 2. Circadian Rhythms and Hypothalamic Control

The hypothalamus is a key target of SCN and the core structure complying with circadian variations in the control of sleep/wakefulness alternation and hormone release, as well as in the regulation of food intake and liver, pancreas, kidney, and heart function [5,6,8,10,24,25,26,27]. A major input to the hypothalamus is about light/dark conditions and temperature. The retinal ganglion cells innervating the SCN are intrinsically light sensitive and mediate in the circadian response to light via melanopsin and through cells in the periventricular hypothalamus and ventral thalamus that are selectively excited or inhibited depending on melanopsin activity [28,29,30]. The hypothalamus also modulates the sensitivity of target endocrine organs for pituitary hormones via its effects on the ANS [31]. The input to hypothalamus regarding temperature is less clear. The first relay nucleus appears localized in the dorsomedial hypothalamic nucleus (DMH) transferring information to the main hypothalamic regulator medial preoptic area (MPA) [32,33]. MPA is also influenced by feeding, metabolism, hormones, inflammation and the time of day, olfactory information from the nucleus of the solitary tract (NTS) and olfactory bulb that trigger an increase in temperature in postprandial time [25,34,35,36].

## 3. Circadian Rhythms and Sleep

The circadian sleep/wakefulness alternation is driven by the SCN, which is reset by light and by melatonin secretion at night [37,38]. Neural networks in the brainstem, hypothalamus, and basal forebrain that selectively activate the cerebral cortex and thalamus are involved. These neurons are inhibited during sleep by GABAergic mechanisms in which the ventrolateral preoptic nucleus (VLPO) seems to have a key role [26]. Galanin- and GABAergic VLPO neurons project during sleep to the hypothalamic and the brainstem nuclei mediating in arousal [39,40,41,42]. VLPO is also important in REM sleep gating via projections on locus coeruleus and dorsal raphe, as well as by histaminergic innervation [43,44,45,46]. Wakefulness is sustained by the ascending arousal system, with the ascending pathway activating the thalamic relays (crucial in information transmission to cerebral cortex) [47] and the lateral hypothalamic area and basal forebrain projecting to the cerebral cortex [48,49,50]. In the first branch, the peduncolopontine and lateral-dorsal tegmental nuclei (PPT/LDT) (cholinergic) provide the main input from the upper brainstem to thalamic-relay nuclei and the reticular nucleus of thalamus [47]. The second branch originates from monoaminergic neurons in the upper brainstem and caudal hypothalamus, including histaminergic tuberomammillary neurons dopaminergic ventral periaqueductal grey matter, locus coeruleus, and raphe nuclei [48,50]. Most of the SCN output projects to the subparaventricular zone (SPZ) and dorsal medial nucleus of the hypothalamus (DMH). The SPZ amplifies the SCN output and projects in the DMH, which in turn mediates in the SCN-controlled sleep-wakefulness regulatory system and projects to VLPO and orexin neurons (orexin is a neuropeptide also known as hypocretin, that regulate arousal, wakefulness and appetite) [51,52,53,54,55,56]. DMH projects to VLPO, LC, and orexin neurons in the lateral hypothalamus [52]. Links between SCN and LC (including DMH, paraventricular hypothalamic nuclei (PVN) and medial and ventrolateral pre-optic areas) are proposed and would confirm the role of the DMH relays in this circuit [52,57]. The selective lesion of DMH suppresses circadian changes in LC activity and reduces circadian rhythms [52,57].

## 4. Circadian Rhythms, Sleep, Homeostasis, and Allostasis

Sleep homeostatic adaptation is described by the accumulation of a *need to sleep* during prolonged wakefulness, with the NREM sleep and REM sleep being regulated by different homeostatic mechanisms possibly mediated for by adenosine [58,59,60,61]. Depletion of adenosine triphosphate and its degradation into adenosine diphosphate and monophosphate results in prolonged wakefulness together with exhaustion of brain glycogen [60]; adenosine activates the VLPO to trigger a sleep episode [59]. The homeostatic and circadian drive for sleep can be temporarily compensated for (as can be the case in emergencies) by shifting to a condition of allostasis (i.e., relative stability during adaptive variation, or in response to a challenge) [62]) through inputs to SNC, VLPO, and orexin neurons from corticolimbic sites (infralimbic cortex, ventral subiculum, lateral septum, and bed nucleus of stria terminalis) [63,64].

## 5. Circadian Rhythms and Age

The SCN clock develops and is responsive to light early in gestation; the retino-hypothalamic tract is identifiable in 36-week human newborns and an effect of the mother’s melatonin is suggested [28,29,30,65]. The day-night cycle in hormone secretion develops between 1 and 3 months, with cyclic cortisol production already at 6 weeks and rhythmical secretion of melatonin beginning at 6 to 12 weeks. The mature core body temperature rhythm and peripheral histone gene expression H3f3b (an indicator of clock function) appear at 6 to 16 weeks; a fully developed circadian rhythm emerges 3 months after full-term birth with maturation of the nocturnal sleep organization; at around 9 months REM sleep decreases and NREM sleep increases during extensive networks remodeling as to dendritic arborization, synaptogenesis, myelinization, neurotransmitter development and programmed cell death. The interactions between brainstem and the thalamic and cortical structures increases [66,67,68,69].

Physiological aging is associated with significant changes in sleep, with a robust tendency toward an evening chronotype. The alignment between sleep and the circadian phase is progressively lost with aging; the circadian phase advances, the circadian rhythm amplitude is reduced, and its period shortened. In the absence of sleep disorders, the circadian markers of the sleep-wakefulness cycle (sleep onset and offset), melatonin (onset), core body temperature (acrophase), and cortisol (acrophase) indicate anticipation of sleep compared to young adults, with earlier bedtime and wake-up. Orexin neurons and their projections appear altered and orexin expression declines with aging [70]. Animal studies agree in indicating that the activity of SCN nuclei governing the circadian system is reduced with aging and the clock gene expression is altered, with age-related degradation at the network level and a decreased effectiveness of peripheral oscillators [71,72,73]. The SCN neuronal population is reportedly unchanged with aging, but significantly fewer vasopressinergic cells are involved in the downstream signaling in elderly rats [73]. Disordered sleep is common among the elderly, who in general report less total sleep time, poorer sleep efficiency, frequent awaking at night, excessive sleepiness during the day, and poor sleep adaptation to adverse circadian phases and sleep-wakefulness misalignment [74,75,76,77,78,79]. The effects of aging on the circadian organization and on each of the CNS and ANS functions conceivably outsize those on CNS/ANS balance, which remains poorly investigated. The time disarrangement over the 24 hrs. between the circadian rhythms (particularly the sleep-wakefulness cycle) and the daily activities impairs physiological adaptation [80,81] to e.g., occasionally increase the incidence (with peaks in morning hours) of severe adverse cardiovascular events such as myocardial infarction, sudden cardiac death, or stroke [82].

## 6. CNS/ANS Interaction and the Central Autonomic Network

Heart rate, respiration, hormone secretion and transport to receptors, smooth muscles activity, and biological sensors are governed by the autonomic nervous system in the process of maintaining homeostatic balance [83]. ANS is also involved in the regulation of innate immune responses and inflammation through neuroendocrine mechanisms and qualifies as a centrally integrated neural reflex [84,85]. The sympathetic (*fight or flight*) and parasympathetic (*rest and digest*) ANS subsystems can act independently or antagonistically, e.g., in modulating the functional balance through inputs from thermoregulation, baroreceptors, chemoreceptors, renin-angiotensin-aldosterone, and atrial and ventricular receptors [86,87,88,89,90]. The functional CNS/ANS interaction involves the brainstem solitary tract nucleus and forebrain structures such as the anterior cingulate, insula, ventromedial prefrontal cortex, amygdala, and hypothalamus with bidirectional interactions between rostral and caudal systems. Parabrachial nucleus/locus coeruleus, cerebellum, periaqueductal gray, hypothalamus, amygdala, and insular and dorsomedial prefrontal cortices structures were found to interact in animal research and human fMRI studies. Some brain regions (dorsolateral prefrontal cortex, mediodorsal thalamus, hippocampus, caudate, septal nucleus, and middle temporal gyrus) have been proposed as possibly unique to humans [51,83,91,92,93,94,95,96,97].

Neural structures and heart function also appear to be peculiarly linked in affective, “cognitive” and autonomic modulation [94,98,99]. A concept model (the Central Autonomic Network; CAN) has been proposed to describe the brain-heart two way interaction [94,96,99]. Several CNS structures are integrated in this model. In particular, the insular cortex, ventromedial prefrontal cortex, anterior cingulate cortex, and amygdala contribute to the modulation of cardiac sympathetic/ parasympathetic outflow. These stractues allow the integration of cardiovascular responses related to behavior and emotion through the hypothalamic area, the parabrachial region of the dorsolateral pons, nucleus of the solitary tract, nucleus ambiguous, rostral ventrolateral medulla, and intermediolateral cell columns [96,98]. The medial prefrontal cortex projecting to interconnected limbic cortical regions and subcortical nuclei (insula, medial temporal lobes, amygdala, ventral striatum, ventral caudate and putamen, mediodorsal thalamus, and hypothalamus) is thought to be a key structure in the brain control of cardiovascular responses [100,101,102].

Heart rate variability metrics (HRV) reflect the complex interaction between brain and the cardiovascular system and describes the ANS functional setup; evidence suggests these measures also reflect (to some extent and indirectly) higher brain functions, and are reliable independent indicators of CNS/ANS interaction [83,97,103,104,105,106,107,108]. HRV, the physiological variation in time of the interval between consecutive heartbeats, is analyzed in the time domain or frequency domains and by non-linear approaches [83,108,109]. The standard deviation of the EKG averaged RR interval reflects the contribution of both the sympathetic and parasympathetic system, while the root mean square is related to the parasympathetic activity. ANS functional contributions can be differentiated by analyses in the frequency domain, usually in three frequency bands: high frequency (HF) (0.15–0.5 Hz), low frequency (LF) (0.04–0.15 Hz), and very low frequency (0.0033–0.04 Hz) (VLF). Recent improvements in non-linear theories provide additional tools to analyze the entropy domain, better describe complexity, and the non-linear temporal relationships with other metrics such as functional connectivity, and extract information [110,111,112,113].

HRV reflects the circadian 24-h cycle with a peak during the second half of the night [114]. Increased HRV during the night relates to the increase in parasympathetic activity, as well as to the interaction between sympathetic and parasympathetic systems [115]. HRV shows a maximal shift toward the sympathetic system during the morning transition from sleep to wakefulness that parallels cortisol peak levels [116]. An increase in heart rate is found around awakening with peak values between 10.00 and 12.00 am (acrophase) and a smaller afternoon peak, after which heart rate begins to decrease to remain lower during the night [117,118]. Other cardiovascular parameters such as blood pressure, cardiac output, and catecholamine serum level are also cyclic [119,120]. A circadian rhythm comparable to those of heart rate and blood pressure has been observed in the activity of the renin-angiotensin system regulating cardiovascular functions [121,122].

## 7. Comments

A complex and pervasive circadian arrangement governs and modulates brain and body functions, the ANS balance, and the CNS/ANS interaction by complex neural networks and non-neuronal factors which interact with each other and depend on circadian timing [123,124,125,126]. Examples in this regard are the sympathetic activation associated with peak cortisol levels in the morning sleep-to-wake transition [116,127,128], and the circadian occurrence of the neurological indices of responsiveness currently used in the classification and prediction of outcome of disorders of consciousness (DoC) [129,130,131,132,133]. The circadian functional arrangement can be disordered by pathologies or temporally disarranged by adaptation to unsuitable conditions, especially if affecting the sleep-wakefulness alternation. Major sleep disorders, inadequate sleeping, and misalignment between human activities and circadian rhythms can result in severe detrimental effects and pathophysiological conditions [63,69,78,80,81,82,134,135,136]. CNS and ANS setups change over time, spontaneously (e.g., to comply with circadian rhythms) or due to homeostatic or allostatic requirements, with different timing and latencies; the CNS/ANS interaction can change accordingly, but with timing difficult to predict and a relationship with the sleep/wakefulness alternation still to be investigated in full detail. HRV only indirectly provides information about the CNA/ANS function and balance. HRV measures at rest and in response to stimulus conditions, however, have high time resolution and can reflect rapid changes better than clinical or neuroimaging markers of damage, also describing the variability during the day and its correlation with the circadian cycle [131,132,133,137,138,139,140].

## 8. Conclusions

Time and the circadian cycle appear to be sources of variability adding to the variety of environmental factors, and may become crucial when considering the ANS major role in internal environment constancy and adaptation that are fundamental to homeostasis. In contrast with experimental and clinical evidence and the availability of tools for noninvasive measurements ntwithstanding, the CNS/ANS interaction has not yet obtained its deserved attention and systematic investigation remains overdue.

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
