# Peer review of "Circadian Rhythms and Measures of CNS/Autonomic Interaction"

_ijerph, 2019, doi:10.3390/ijerph16132336_

Round 1

Reviewer 1 Report

The article presents a review about the interaction between the circadian rhythms and autonomic regulation. However, the article lacks clear objectives, which make it difficult to understand why the paper has been written.  For instance, what does the paragraph “circadian rhythms and age” provide for the purpose of the paper? Moreover, there is a lack of precision in the description of the organization of the circadian system and in the sleep regulation structures.

In my opinion, the authors should rewrite the article, considering the aims of the journal and specifying clearly what is the purpose and objectives of this revision.

Also, authors would like to revise the bibliography and use the necessary and actualized references. Just to mention one exemple, the groups of Buijs and Kalsbeek have widely published about ANS and circadian control, and here only two references, from 2001 and 2003, are shown.

Author Response

The referee's comments and suggestions are legitimate and correct. The paper could be reformatted and possibly improved. However, it has been organized to meet the special issue project and in agreement with the guest editor. We will consider re-writing the paper if the guest editor agrees and extra time can be allowed. Some editing has been done anyway, and we hope it may be useful.

Reviewer 2 Report

The authors provide a very informative collection of concepts and articles dealing with a difficult and novel topic. This article would be of interest to a wide variety of researchers and a useful resource.

It is very clear organized and easy to read. Here are my comments by line:

40 Local clocks in tissues are also called peripheral clocks. ...Later in the article the term peripheral clock is used, so good to introduce the concept and terminology in line 40.

41 Determines the 'sleep structure' (does this refer to slow wave REM phases, if so does SCN determine this?). ...The article explains it more later on, may be good to introduce earlier.

Unclear how temperature fits in- does temperature fine tune circadian rhythm, or other way around? Light and temperature seem to be mentioned as similar concepts of entrainment

65 double verb: active projecting

78, and 170 extra space

79 where are the lesions located - may be interesting to elaborate more

93- how is light received by developing fetus

118 some unformatted citations appear

129 SNS and PNS are not necessarily antagonistic of each other - classically they are known to be antagonistic for heart rate but for some functions like urination or sweating they can be co-operative. Many functions are affected by one but not the other.

114 HRV acronym defined twice

157 standard deviation does not require capitalization

171 Increase > increase

174 as > such as

174  This part is unclear,  "show  higher  level during the night and falling during the night" does it refer to a rhythm, or is it respectively applying to the various parameters listed earlier in the sentence.

184 DoC not defined

191 provide > provides

other thoughts:  gastrointestinal  motility is mentioned early in intro but is not mentioned again

allostasis is perhaps not such a common term, define for reader, how does it differ from homeostasis.

Orexin is fairly new conceptually could be introduced a bit more

The term 'chronobiology' could be used in intro to describe the field

Eating meals is another important regulator and cause for entrainment of rhythms (and in turn affected by rhythms).

Author Response

All suggestions have been incorporated in the revised ms. Thank you for helping.

Reviewer 3 Report

The manuscript by Riganello et al. presents a review of the literature on circadian rhythm and the interactions between central  and autonomic nervous systems.

The subject is of clear interest, the number of references is adequate and the manuscript is enjoyable. I have few suggestions to be considered:

1. I would explain acronyms when first used, even if common, for increasing accessibility to broader public ( for example “mo.”, “wks” etc)

2. Considering the intricacy of the subject, I would suggest a figure for summarising and explaining the key points.

Author Response

All suggestions have been incorporated in the revised ms. Thank you for helping. Two figures (at least) would be necessary to outline the matter; it will be done if extra time is allowed before the special issue is due for printing.

Round 2

Reviewer 1 Report

The authors have modified some editing and some new references have been added to the new version of the manuscript.